# Effects of Multi-Ingredient Preworkout Supplementation across a Five-Day Resistance and Endurance Training Microcycle in Middle-Aged Adults

**DOI:** 10.3390/nu12123778

**Published:** 2020-12-09

**Authors:** Joel Puente-Fernández, Marcos Seijo, Eneko Larumbe-Zabala, Alfonso Jiménez, Gary Liguori, Claire J. L. Rossato, Xian Mayo, Fernando Naclerio

**Affiliations:** 1Institute for Lifecourse Development, School of Human Sciences, Centre for Chronic Illness and Ageing, University of Greenwich, Eltham SE9 2TB, UK; J.PuenteFernndez@greenwich.ac.uk (J.P.-F.); M.Seijo@greenwich.ac.uk (M.S.); C.Rossato@greenwich.ac.uk (C.J.L.R.); 2School of Doctorate and Research, European University of Madrid, 28670 Villaviciosa de Odon, Spain; eneko@cop.es; 3Advanced Well-Being Research Centre, Sheffield Hallam University, Sheffield S1 1WB, UK; alfonso.jimenez@ingesport.es; 4Observatory of Healthy and Active Living of Spain Active Foundation, Centre for Sport Studies, King Juan Carlos University, 28942 Madrid, Spain; 5GO Fit LAB, Ingesport, 28108 Madrid, Spain; xian.mayo@urjc.es; 6College of Health Sciences, University of Rhode Island, Kingston, RI 02881, USA; gliguori@uri.edu

**Keywords:** multi-nutrient, preworkout supplement, tensiomyography, performance, dietary supplements

## Abstract

Preworkout multi-ingredient admixtures are used to maximise exercise performance. The present double-blind, cross-over study compared the acute effects of ingesting a preworkout multi-ingredient (PREW) admixture vs. carbohydrate (CHO) over a week (microcycle) comprising three resistance training (RT) workouts alternated with two 30-min low-intensity endurance sessions (END) on RT volume (kg lifted) and END substrate oxidation. Additionally, postworkout decreases of muscle function and subjective responses were analysed. Following a baseline assessment, fourteen recreationally trained, middle-aged adults (seven females, 48.8 ± 4.7 years old) completed two identical microcycles separated by a two-week washout period while receiving either PREW or CHO (15 min prior to workout). The RT volume, per session (SVOL) and for the entire week (WVOL), was calculated. Fatty acid oxidation (FAO) during 30-min cycling corresponding to their individually determined maximal fat oxidation was measured using expired gasses and indirect calorimetry. Assessments of performance and tensiomyography were conducted within 20 min after each RT. Higher (*p* = 0.001) SVOL and WVOL along with a larger proportion of FAO (*p* = 0.05) during the second END workout were determined under the PREW treatment. No other statistically significant differences were observed between conditions. Compared to CHO, a preworkout multi-ingredient appears to increase resistance volume and favour fat oxidation during low-intensity endurance exercises.

## 1. Introduction

Multi-ingredient preworkout admixtures have been proposed as a specialised category of dietary supplements to be administered prior to exercise with the aims of increasing motivation to train and maximising exercise performance outcomes [1]. Previous studies on preworkout multi-ingredient admixtures have shown several physical performance-related benefits such as (i) improving muscular endurance [2], (ii) increasing peak power output [3], (iii) higher training volume capacity [4], and (iv) greater fatty acid oxidation [5]. Furthermore, additional effects on subjective feelings (e.g., increased perceived alertness and focus on the task, and improved energy levels) have been reported [6].

Most preworkout multi-ingredient admixtures contain a proprietary blend of ingredients claiming to produce performance benefits when taken as described. Several preworkout admixtures contain caffeine mixed with other ingredients in an attempt to produce a synergistic, ergogenic effect that favours fat metabolism during long-lasting activities [7]. Additionally, the combination of ingredients such as caffeine and amino acids (e.g., L-Citrulline DL-Malate, L-Tyrosine, L-Taurine, etc.) has been shown to delay fatigue [8] and improve the overall resistance training volume [4]. The ergogenic effect of caffeine on exercise performance has been mainly attributed to its action as an adenosine receptor blocker [9], which may serve to decrease the perception of fatigue [10], stimulate the release of excitatory neurotransmitters, and therefore, increase motor neuron excitability [11]. These effects of caffeine may therefore speed up cell’s energetic demands and prolong lipolysis [12].

Combining caffeine with yerba mate containing caffeoyl derivatives (chlorogenic acid, phytosterols and saponins) promoted fat metabolism [13], increased fatty acid oxidation, and reduced the perception of effort during endurance exercises performed at the intensity associated with the highest fat oxidation [5]. Additionally, preworkout admixtures including high-quality protein with added amino acids (e.g., L-leucine, L-arginine, L-tyrosine, or L-taurine) or derivatives (e.g., citrulline-malate, betaine or L-carnitine) may augment the effects of caffeine by increasing muscle efficiency and delaying fatigue [14]. L-taurine is an amino-containing sulfonic acid; its inclusion as an ingredient in preworkout multi-ingredient admixtures has been associated with higher muscular endurance capacity during resistance exercise [7]. Betaine is another naturally-occurring derivative of the amino acid glycine which has been shown to increase muscle blood flow by elevating the levels of nitric oxide and promoting fluid and thermal homeostasis [7]. L-Carnitine is a conditionally essential amino acid-like molecule found predominantly in skeletal muscle with an essential role in fatty acid metabolism [15]. Preworkout L-carnitine supplementation has been associated with increasing vasodilation, favouring oxygen supply, and attenuating exercise-induced hypoxia [16]. Additionally, combined with L-leucine, oral L-carnitine supplementation can prevent protein catabolism by stimulating mTOR expression [17]. Isomaltulose is a disaccharide comprised of glucose and fructose which has been shown to slow rates of hydrolysis and subsequent absorption at the intestinal mucosa, resulting in prolonged glucose delivery to the systemic circulation. Its ingestion before intermittent exercises favoured a more stable glycaemic response in athletes [18].

It is possible that a combination of both caffeine and yerba mate extract with proteins and slow-release carbohydrate (isomaltulose) admixtures including citrulline-malate, L-leucine, L-tyrosine L-taurine, and betaine may work synergistically to acutely enhance performance beyond what is possible with any one single ingredient.

To the best of the authors’ knowledge, no previous studies have investigated the potential ergogenic effect of a similar admixture administered before a workout. Furthermore, a meta-analysis of 35 trials concluded that combining multi-ingredients with resistance training is an effective strategy to induce greater gains of fat-free mass and strength [19]; however, the observed changes were more evident in untrained and elderly individuals (>45 years; 66 ± 8 years) compared to their trained and younger counterparts [19]. The aforementioned meta-analysis did not distinguish among studies using different administration protocols, e.g., pre- vs. post- workout or daily multidose intakes. Indeed, none of the included trials conducted with middle-aged and older adult participants considered a preworkout supplementation alone.

The aim of the present study, therefore, was to compare the acute effects of ingesting a preworkout multi-ingredient (PREW) admixture over a training week (microcycle) including both resistance and endurance training vs. an isocaloric preworkout placebo containing carbohydrate alone (CHO) in recreationally trained middle-aged men and women on (i) resistance training performance, (ii) substrate oxidation during endurance exercise, (iii) postworkout decrease of muscle function, and (vi) subjective measures. The primary outcomes were: (i) the total resistance training volume performed per workout (SVOL) and for the whole week (WVOL), measured in kilograms of lifted load in each exercise and normalised by the fat-free mass, and (ii) the amount of fat and carbohydrate oxidised during the endurance sessions. Secondary outcomes were the estimated decrease of muscle function due to the performed RT on (i) medicine ball throw distance, (ii) jump height, (iii) isometric strength, (iv) the evoked tensiomyography (TMG) contraction velocity (Vc) of vastus medialis (VM), biceps femoris long head (BFLH), and anterior deltoids (AD). In addition, we sought to describe the effects of preworkout supplementation on the following exploratory variables: (i) subjective feelings of energy, focus and awareness on the task, (ii) the postworkout perceived exertion, and (iii) the rest of the TMG variables [maximum radial muscle displacement (Dm), contraction time (Tc)].

We hypothesised that compared to the ingestion of carbohydrate alone, a preworkout multi-ingredient admixture will promote a higher resistance training volume and favour a greater proportion of fatty acid oxidation during endurance exercises. Additionally, it will attenuate the decrease of muscular function after resistance workouts.

## 2. Materials and Methods

Following inclusion, familiarisation, baseline assessments, and a five-day recovery period, using a randomised, counterbalanced, cross-over, double-blind, placebo-controlled design, the participants were randomly allocated to receive either PREW or CHO. Thereafter, the participants completed two identical training and testing microcycle periods (five days each) separated by a two-week washout period. The nutritional treatment was switched from the first to the second five-day training and testing period (Figure 1).

### 2.1. Participants

Fourteen recreationally active, middle-aged adults (seven females) participated in this study. To be eligible, participants were required to have been training regularly two to three times per week, using routines including resistance exercises (e.g., bench press, leg press, squats, or lunges) for a minimum of six months before the beginning of the study. Exclusion criteria included anyone suffering from recent (last six months) or present injuries which may prevent them from performing the required exercises, suffering from current illnesses or chronic diseases (including metabolic syndrome, advanced obesity, or sarcopenia), or taking any medication or supplements that would affect exercise performance (i.e., protein amino-acids supplements, NSAIDs, etc.). All female participants were premenopausal and were randomly tested throughout their menstrual cycle [20]. All participants provided written informed consent in accordance with the Declaration of Helsinki. Procedures were approved by the University of Greenwich Research Ethics committee (FES-FREC-18-3.04.16) on 23 January 2020. The project was registered as a clinical trial at the U.S. National Institutes of Health. https://www.clinicaltrials.gov (NCT041477741).

To determine the appropriate sample size, an interim analysis was performed once six participants had completed the study. The analysis was conducted based on the most relevant primary outcome measure [the WVOL, summarizing the total load from the performed exercise over the three conducted resistance training sessions]. Assuming an α-error of 0.05, for the resulting effect size of d = 0.96 calculated between two dependent means determined for the PREW and CHO conditions, the required sample size of n = 11 was estimated to achieve >80% statistical power.

### 2.2. Procedures

#### 2.2.1. Familiarization

After confirming their eligibility for the study, the participants completed three sessions of familiarization (week 1) during which the training protocol, exercise techniques, and the assessment procedures were explained.

#### 2.2.2. Baseline Assessments

Environmental conditions kept constant in all testing and training sessions, i.e., mean (SD): 20 (1) °C, 775.6 (12) mmHg and 51.1 (6.1) % for air temperature, barometric pressure, and relative humidity, respectively. Participants refrained from heavy exercise during 48-h prior to all baseline assessments. Tests were conducted within one day and in the following order: (i) body composition, (ii) tensiomyography, (iii) medicine ball throw, (iii) vertical jump, (iv) maximal isometric mid-thigh pull, and (v) incremental cycling test to exhaustion. A passive 10-min recovery period was provided between each individual test.

#### 2.2.3. Body Composition

The standard measurements were performed following the recommendations for anthropometric assessment [21]. To eliminate interobserver variability, one investigator consistently performed all measurements. Height was measured in a stretched stature to the nearest 0.01 m using a wall-mounted stadiometer (Seca GmbH, Hamburg, Germany) and body mass (BM) was corrected to the nearest 0.1 kg using a digital scale (Seca GmbH, Hamburg, Germany). Fat mass (FM) and fat-free mass (FFM) were estimated from whole-body densitometry using air displacement via Bod Pod^®^ (Life Measurements, Concord, CA, USA) and following the manufacturer’s instructions as detailed elsewhere [22].

#### 2.2.4. Medicine Ball Throw (MBT)

Three overhead throws were performed using the methodology described by Viitasalo [23]. Based on the distance, the best of three attempts was chosen for the analysis. Males used a five-kg (circumference 0.30 m), and females a three-kg (circumference 0.21 m) medicine ball.

#### 2.2.5. Vertical Jump

A countermovement jump (CMJ) was performed according to the methodology described by Brown and Weir [24]. A Kistler force platform (9287B, three component force platform; Kistler, Hook, United Kingdom; dimensions: 900 × 600 × 100 mm) with a sampling rate of 2000 Hz was used to calculate the height from the difference, in meters (m), between maximum height of the centre of mass (apex) and the last contact of the toe on the ground during the take-off. Based on the height, the best of three jumps was chosen for the analysis.

#### 2.2.6. Maximal Isometric Force (MIF)

A T.K.K. 5402 dynamometer (Takei Scientific Instruments Co. Ltd., Niigata, Japan) with a base of 31.5 × 31.5 cm, chain (51 cm) and latissimus pulldown bar (120 cm; Perform Better, United Kingdom) was used to assess full-body MIF [25]. The participants were instructed to adopt a position similar to the second pull in the power clean exercise (mid-thigh pull). Participants were positioned by standing on the foot grips and adjusting the chain length to have the bar positioned slightly above the knees. Participants gripped the bar without straps, and before pulling, maintained tension on the chain to avoid jerking movements. Thereafter, participants pulled upwards using as much force as possible [26]. Three attempts of 5 s with 30 s rest were conducted. The maximum recorded value in kilograms force (kgF) was selected for the analysis.

#### 2.2.7. Tensiomyography Assessment

A TMG portable device (TMG Measurement System, 146 TMG-BMC Ltd., Ljubljana, Slovenia) with a maximal stimulation output of 110 mA·ms^−1^ was used to measure the contractile properties of the Vastus Medialis (VM), Biceps Femoris Long Head (BFLH) and Anterior Deltoid (AD) at the dominant limb [27,28]. Measurements were collected by the same trained researcher, following the methodology described by Rey et al. [29] and obtained at rest, in supine position for the VM, prone position for BFLH and sitting position for the AD. Changes in the evoked muscular contractile properties were estimated by analysing the following variables (i) maximal radial displacement of the muscle belly (Dm), contraction time between 10 and 90% Dm (Tc), and mean velocity of contraction (Vc), which was calculated by dividing the Dm by the sum of the Tc and the delayed time (Td) [27]. These three variables demonstrated high levels of accuracy, reliability, and sensitivity to changes in neuromuscular function by TMG analysis [30]. Furthermore, it is not uncommon for Tc and Dm to vary disproportionately relative to one another, and changes in Tc, independent from Dm, can be due to alterations in the rate of contraction, as measured by Vc [31].

The intraclass correlation coefficients (ICC) at 95% confidence intervals (CI) for TMG variables ranged from 0.88 to 0.91, similar to those reported in previous investigations [28].

#### 2.2.8. Incremental Cycling Test to Exhaustion

Following a standardised warm-up, participants completed a maximal incremental laboratory exercise test to exhaustion on a cyclo-ergometer (Lode Corival^®^). The test commenced at a work rate of 30 or 50 W (women and men, respectively) and increased 15 W every minute. The participants were encouraged to keep a constant cycling rate between 60 and 90 rpm while remaining in a sitting position. When cadence dropped by more than 10 rev·min^−1^ for more than 10 s despite strong verbal encouragement, tests were terminated. The test was designed to avoid long-term muscular fatigue, and every trial lasted <18 min. Expired gases were collected continuously during the test using a Cortex MetaLyzer 3B gas analyzer (Cortex Biophysik, Leipzig, Germany). This device was also used to calculate the respiratory exchange ratio (RER), and thereafter, to estimate the oxidation and relative contributions of carbohydrate and fat across the test [5]. Additionally, the maximum heart rate (HRmax) and VO_2_peak (calculated as the highest mean oxygen consumption over a 30-s period [32]) were calculated for descriptive purposes.

### 2.3. Dietary and Supplementation

Each participant completed a three-day food diary report (two weekdays, and one weekend day). The Food Processor Software (Version 11.4.70, London, UK) was used to calculate the energy and nutritional compositions of the reported diets. Participants were instructed to maintain their habitual diet throughout the study, including the washout period. They were asked to report any minimal change regarding food composition and size, ingestion of supplements or compliance with the reported meals, including breakfast, lunch, pre- and post- workout food intake, and dinner. If any change had been detected (i.e., becoming vegetarian, restricting calories, taking additional nutritional supplements, etc.), that participant’s data would have been excluded from the analysis.

During the five-day training periods (weeks 3 and 6), all participants consumed either one 40 g dose of a commercially available preworkout multi-ingredient admixture (PreWorkout PRO ST, Crown Sport Nutrition, Spain) or an isoenergetic placebo (see Table 1).

The two products under study were presented in sachets of vanilla-flavoured powder to be diluted in ~350 mL of cold plain water and administered 15 min before each workout-session. The diluted drinks were similar in appearance, texture, and taste. The participants were instructed to ingest the last meal ~3 to 4 h before each training session. They were allowed to drink water but not to ingest any food during the 3-h preworkout time or after completion of all postworkout assessments. An investigator who was not involved in the data collection prepared and administered both supplements for all participants, providing double-blinding of both the participants and the data collection researcher. No supplements were consumed on non-exercising days (e.g., weekends and weeks 4 and 5). Furthermore, to avoid possible confounding trial order effects, the conditions were tested following a balanced randomised order. Following the preintervention assessments, participants were matched by sex and body mass. Assignment of participants to treatments was performed by block randomization using a block size of two and in a double-blind (PREW or CHO) fashion.

### 2.4. Exercise Protocols and Postworkout Assessments

*Resistance Training (RT):* Three RT sessions were conducted on alternate days (e.g., Monday, Wednesday, and Friday), and all sessions took place late in the afternoon (4 to 6 pm). Each participant performed a supervised full-body resistance-training protocol involving a standardised warm-up followed by three circuits of one set of the following exercises: (i) parallel squat, (ii) bench press, (iii) alternate 40 cm box step-ups, (iv) shoulder press, (v) alternate lunges, (vi) upright row, (vii) deadlift, and (viii) squat jumps. About 30 sec rest between exercises and 3 min between circuits was allowed. As the workout aimed to create a high level of mechanical and metabolic stress, a muscle endurance training targeting 16 self-determined maximum repetitions (>40 to <60% 1RM) per set was designed [33]. When participants were able to perform more than 16 repetitions per set, the load was increased between 2.5 and 5 kg. If fewer than 16 repetitions were completed, a minimum rest period of 15 s was introduced until the participants were able to reach the targeted number of repetitions per set. The time to complete the workouts was ~55-min. Additionally, the rate of perceived exertion (RPE) for the entire workout was measured by using the OMNI-RES scale [34]. To avoid easy or difficult elements toward the end of the sessions from skewing the overall rating of the exertion, the participants were asked to rate their session-RPE by answering the question “How hard was your entire workout?” between 15 to 30 min after the completion of each resistance training workout [35]. Within 20 min after the completion of each workout (RT1, RT2 and RT3), assessments of voluntary (MBT, CMJ, MIF) and evoked (TMG) muscular function were conducted, following the same protocol as used for the baseline assessment.

*Endurance Training (END):* Two END sessions were conducted twice a week on alternate days (e.g., Tuesday and Thursday). Each session involved 30 min of cycling at an individually determined maximum fatty acid consumption (Fatmax) intensity. Additionally, the RPE using the Borg scale (6–20) was measured every five minutes during exercise. The exercise intensity at which the reliance on fat oxidation reached its maximum (Fatmax) was individually determined and chosen as the target intensity for the endurance training sessions. The metabolic data of fat (FAO) and carbohydrate (CHOox) oxidation were estimated using stoichiometric indirect calorimetry equations (Equations (1) and (2)), assuming minimal protein contribution during exercise.
(1)$$\mathrm{FAO}\;=\;1.695\;\times \;\dot{V}O_{2}\;-\;1.701\;\times \;\dot{V}CO_{2}$$
(2)$$\mathrm{CHOox}\;=\;4.585\;\times \;\dot{V}CO_{2}-\;3.226\;\times \;\dot{V}O_{2}$$

The FAO during exercise was determined based on averaging the last twenty min of the 30-min session. Fatmax (g·min^−1^) was determined from the incremental test as the highest amount of FAO averaged over 30-s. Fatmax corresponding intensity was determined as the mechanical power output (W) and relative intensity (%) relative to peak power (P_peak_), at which each participant achieved Fatmax [5].

Subjective Feelings: Prior to each workout session and after taking the supplement, all participants were asked to complete a four-question questionnaire based on a five-point rating scale. The participants were asked to rate their energy level, fatigue level, feelings of alertness, and feelings of focus on the task using the following verbal anchors: 1 = very low, 2 = low, 3 = average, 4 = high, and 5 = very high, with the average response of the three testing sessions computed for a final “score” [6]. The same researcher performed all test administrations under controlled conditions (i.e., in a quiet room).

Compliance with the study procedures (e.g., potential changes in dietary intake, ingestion of caffeine or other supplement, resting and training time) were checked by the researchers using an individual interview before starting all training or testing sessions.

### 2.5. Statistical Analyses

Descriptive analyses were performed, and Shapiro-Francia tests were applied to assess normality. Before testing the main hypothesis, the possible treatment order effect, and the effectiveness of the washout phase to rule out any carryover effect was checked. For all the analysed variables, a preliminary test using the sum of all values obtained for each participant at any training sessions (RT1, END1, RT2, END2, RT3) in the two periods was calculated and compared across the two sequenced conditions. We used an independent samples Student’s T-test to compare the values measured in the seven participants who started with PREW vs. the results determined for the seven others who started with CHO [36].

Two-way repeated-measure ANOVA (3 RT workouts or two END workouts × two conditions (PREW vs. CHO)) was performed to respectively analyse (i) the SVOL lifted and the session-RPE rated in each RT workout, and (ii) the substrate oxidation (FAO and CHOox) and the averaged Borg-scale score measured during each END session.

A related sample Students T-test was used to analyse differences between conditions (PREW vs. CHO) for the WVOL and the averaged measures of the four questions included in the subjective feelings questionnaire.

Raw changes in performance (MBT, CMJ and MIF) and TMG, were calculated by subtracting pre- from post- assessment values, without adjusting for pre-values, since the same participants performed under both conditions acting as their own controls. In order to assess the magnitude of the differences from baseline, confidence intervals (CIs) of the differences were calculated and plotted. Those CIs not crossing zero were considered statistically significant from the baseline performance. Additionally, two-tailed one-sample Student T-tests were used to test for a null effect hypothesis.

To compare differences between conditions (PREW vs. CHO) at postworkout measurements in raw change, an ANOVA with repeated measures was used to examine changes over the three sessions (RT1, RT2, and RT3) for MBT, CMJ, MIF, VOL, GRPE (global rate of perceived exertion after 20 min of the end of the workout), and all TMG variables. Differences over time were compared using Bonferroni-adjusted pairwise comparisons when appropriate.

A previous analysis using sex as an interparticipant factor (i.e., condition × time × sex) demonstrated no significant interactions between sex and conditions or times for all the variables with the exception of TMG. Therefore, sex (men, women) was used as a covariate to analyse changes in the TMG variables. For the rest of the variables, data were pooled between sexes and analysed together for the rest of the variables.

Eta squared (*η*^2^) and Cohen’s d values were reported to provide an estimate of the standardised effect size (small *η*^2^ = 0.01, d = 0.2; moderate *η*^2^ = 0.06, d = 0.5; and large *η*^2^ = 0.14, d = 0.8). The significance level was set at 0.05. All results are reported as mean (standard deviation) unless stated otherwise. All statistics were performed using the Statistical Package for the Social Sciences (SPSS for Windows, version 26.0; SPSS, Inc., Chicago, IL, USA).

## 3. Results

The demographic characteristics of the sample are presented in Table 2.

No carryover effect was observed for the main performance (WVOL *p* = 0.40, SVOL *p* = 0.51) and metabolic (FAO *p* = 0.39, CHOox *p* = 0.77) variables, nor for the secondary variables (MBT *p* = 0.43, CMJ *p* = 0.21, MIF *p* = 0.37, Vc for VM *p* = 0.36, BFLH *p* = 0.51 and AD *p* = 0.10) and the exploratory variables (all *p* > 0.05).

### 3.1. Diet Analysis

Table 3 shows the daily consumption of macronutrients (grams) and energy (kcal) including and not including the two analysed supplements.

Overall, the ingestion of a 40 g daily dose of PREW increased protein and carbohydrate intake, while adding 27 g of maltodextrin increased daily carbohydrate ingestion alone. Both PREW and MALT significantly increased energy intake. 

### 3.2. Primary Outcomes

*Total Resistance Training Volume over the entire week (WVOL):* Under the PREW condition, the participants lifted more kg (*p* = 0.001, d = 1.26) than when they ingested only maltodextrin (Figure 2A).

#### 3.2.1. Total Resistance Training Volume Per Session (SVOL)

Effects between workouts (F[2,26] = 15.082, *p* = 0.001, η^2^ = 0.010) and supplements (F[2,26] = 22.295, *p* = 0.001, η^2^ = 0.023) were observed but an interaction was not found (F[2,26] = 1.073, *p* = 0.375, η^2^ = 0.001. Post hoc comparisons indicated that under PREW, the participants lifted more kilograms than under CHO (*p* < 0.05, d > 0.80) during the three RT sessions (Figure 2B). Additionally, under PREW, the training volume was significantly higher in RT3 vs. RT2 (*p* = 0.026) and RT1 (*p* = 0.005), as well as between RT2 vs. RT1 (*p* = 0.016). However, under CHO, no difference was observed between the SVOL during the first two sessions (RT1 vs. RT2, *p* = 0.471), while RT3 increased significantly compared to both RT1 (*p* = 0.008) and RT2 (*p* = 0.023).

#### 3.2.2. Subtract Oxidation during Endurance Training Sessions

Main effects between supplements (F[1,13] = 4.878, *p* = 0.046, η^2^ = 0.077), but not between workouts F[1,13] = 0.008, *p* = 0.931, η^2^ = 0.001) or interaction F[1,13] = 1.256, *p* = 0.283, η^2^ = 0.088) effects, were determined for FAO.

No main effects between workouts (F[1,13] = 0.169, *p* = 0.663, η^2^ = 0.001), supplements (F[1,13] =1.922, p = 0.189, η^2^ = 0.007) or interaction (F[1,13] = 3.242, p = 0.095, η^2^ = 0.088) were determined for CHOox.

The post hoc analysis indicated a significantly higher FAO (*p* = 0.05, d = 0.53) and a non-significantly (*p* = 0.07, d = 0.51) lower CHOox under PREW compared to CHO during END 2 (Figure 3).

### 3.3. Secondary and Exploratory Outcomes 

#### 3.3.1. Medicine Ball Throw (MBT)

Compared to baseline, a significant decrease in performance was observed after the completion of RT2 and RT3 under the PREW condition, and after completion of the three RT workouts under the CHO condition. Additionally, the main between supplement effects were determined (F[1,13] = 12.65, p = 0.004, η^2^ = 0.065), though not for workouts (F[2,26] = 1.920, *p* = 0.167, η^2^ = 0.019) or interaction F[2,26] = 1.073, p = 0.357, η^2^ = 0.016). A post hoc analysis revealed significantly lower performance reduction (*p* = 0.001, d = 1.46) under the PREW compared to the CHO condition after RT 2. No between supplement differences were observed after RT1 and RT3 (Figure 4A).

#### 3.3.2. Vertical Jump (CMJ)

Compared to baseline, no significant reduction in VJ performance was measured under either condition (PREW and CHO) after the three RT workouts. No main effects were determined between workouts (F[2,26] = 1.994, *p* = 0.156, η^2^ = 0.011), for supplements (F[1,13] = 0.876, *p* = 0.366, η^2^ = 0.006), or interaction (F[2,26] = 0.651, *p* = 0.530, η^2^ = 0.005) (Figure 4B).

#### 3.3.3. Maximal Isometric Force (MIF)

Compared to baseline, significant strength decreases were observed after RT1 (*p* = 0.006, d = 0.88) and RT2 (*p* = 0.039, d = 0.61) only under CHO condition. No strength reduction was measured under the PREW condition (Figure 4C).

There were main effects between supplements (F[1,13] = 4.881, *p* = 0.046, η^2^ = 0.076) but not for workouts (F[2,26] = 2.854, *p* = 0.076, η^2^ = 0.022) or interaction F[2,26] = 0.858, *p* = 0.436, η^2^ = 0.005). The post hoc analysis revealed a significantly lower performance reduction (*p* = 0.007, d = 0.86) under the PREW compared to the CHO condition after RT 3. No between supplement differences were observed after RT1 and RT2 (Figure 4C).

#### 3.3.4. Tensiomyography

Compared to baseline, no significant changes (all *p* > 0.05) in any TMG variable were observed (Vc, Dm or Tc). No between workouts, supplements, or interaction effects were determined in the three analysed muscles (AD, BFLH and VM). Means and SD describing the changes measured after RT1, RT2, and RT3 of the three TMG analysed variables (Vc, Dm and Tc), including the 95% CI and the ANOVAs tests for the three muscles under the two assessed conditions (PREW and CHO), are presented as Appendix A.

#### 3.3.5. Perception of Effort

The GRPE rated by the OMNI-RES scale after completion of the three RT workouts were as follows: (i) RT1: PREW 7.9 ± 0.9; CHO 7.9 ± 1.2, (ii) RT2: PREW 8.0 ± 0.8; CHO 8 ± 0.8, (iii) RT3: PREW 8.1 ± 0.7; CHO 8.1 ± 0.8. Neither main effects between workouts (F[2,26] = 1.387, *p* = 0.268, η^2^ = 0.010) and supplements (F[1,13] = 0.07, *p* = 0.795, η2 = 0.001) nor an interaction F[2,26] = 0.038, *p* = 0.963, η^2^ = 0.001) were determined.

The average RPE values rated by the Borg-scale (6–20 points) every 5 min during the endurance training were as follows: (i) END1: PREW 11.5 ± 1.1; CHO 12.0 ± 1.2, (ii) END2: PREW = 11.5 ± 1.2; CHO 11.7 ± 1.1. No main between workouts (F[1,13] = 1.230, *p* = 0.287, η^2^ = 0.003), supplements (F[1,13] = 2.149, *p* = 0.166, η^2^ = 0.026) or interaction F[1,13] = 2.997, *p* = 0.107, η^2^ = 0.006) effects were determined.

#### 3.3.6. Subjective Feelings

The responses to the subjective feelings questionnaire are presented as Appendix A. No differences between conditions (all *p* > 0.05) were observed for the averaged energy levels values, fatigue level, feeling of alertness, or focus for the task before starting the workouts.

## 4. Discussion

The observed results suggest that PREW vs. CHO treatment increases the total volume lifted over three resistance training sessions performed across a five-day macrocycle (Figure 2). Additionally, ingesting the multi-ingredient admixture before low-intensity (Fatmax) endurance training sessions favoured fat oxidation (Figure 3A). However, even though participants experienced less performance decrease on the MBT and MIF under the PREW condition after completion of RT2 and RT3 respectively, the lack of interaction effect precludes any conclusion supporting the advantage of PREW over CHO to attenuate this decrease (Figure 4). No differences between supplements (PREW vs. CHO) were identified for the remaining variables, including the acute response of involuntary muscular function (TMG), the rating of perceived effort measured after both RT (by the OMNI-RES scale) and END (by the 6–20, Borg-scale), or the subjective feelings. Based on these findings, and within the confines of the study procedures, we accept our research hypothesis asserting that compared to carbohydrates alone, a preworkout multi-ingredient admixture may promote higher training volumes (kg lifted) during resistance workouts and may favour fatty acid oxidation during submaximal continuous endurance exercises. However, we cannot accept the hypothesis supporting the benefit of a preworkout multi-ingredient admixture instead of carbohydrate alone to attenuate the postworkout decrease of muscular function, improving the perception of effort, and enhancing subjective feelings.

Previous studies reported enhancement effects of ingesting preworkout multi-ingredient admixtures on resistance training performance [4,6,37,38]. Compared to the multi-ingredient admixture examined in our study, the formulations used in the aforementioned investigations provided less energy, containing lower amounts of carbohydrates and proteins, but included creatine, beta-alanine, or both. These two ingredients were not part of the admixture used in our study. However, all the formulations, including the one tested in the present study, contained alkaloids, mainly in the form of anhydrous caffeine, combined with amino-acids or derivatives (e.g., L-Leucine, Taurine, L-Citrulline-DL-malate). The observed increased volume during the three RT workouts under the PREW condition could have been related to the role of caffeine as an adenosine receptor antagonist [39]. Conversely, previous studies reported a positive effect of caffeine to decrease the perception of effort during and after high-intensity exercise sessions [40] and improve task motivation [41], focus on the task, and energy feelings [6] compared to the ingestion of carbohydrate alone, even though we failed to observe such effects. However, it is worth noting that the higher RT volume accomplished under the PREW condition did not elicit a concomitant increase in the global postworkout perception of effort. Therefore, it is possible to speculate that the ingestion of the multi-ingredient admixture attenuated the concomitant rise of the perception of effort that could have been expected by the completion of a higher training volume. Additionally, the increased volume under the PREW condition should have been followed by a larger decrease in voluntary and involuntary muscular function. However, there was no difference between conditions. In practice, the ingestion of the multi-ingredient, in addition, to promote an increased RT volume, attenuated the concomitant loss of both voluntary and evoked muscular function that could have been expected after performing higher volumes during the resistance workout sessions [42,43]. On the other hand, the lack of clear differences between the two tested conditions in all the TMG variables can be explained by the expected blunted-fatigue effect of preworkout supplementation [44]. Nutrients included in both PREW (e.g., isomaltulose and amino acids) and placebo (maltodextrin), to a certain extent, were effective at attenuating the drop in muscle function that is often expected with no nutrient ingestion [45]. In any case, the observed mitigation of usual performance decreases under the PREW condition on both MBT and MIF supports the efficacy of ingesting PREW vs. CHO. The mentioned benefit, however, still needs confirmation from further studies conducted over longer intervention periods (>4 weeks) and using larger samples.

It is worth mentioning that, up to a certain limit, e.g., ten sets per muscle group per week [46], substantially greater training volumes may be beneficial in enhancing muscle hypertrophy [47]. Furthermore, appropriate lean and fat mass levels are two of the most relevant health-related factors to improve the quality of life and attenuate deleterious processes associated with typical ageing [48]. In those lines, multi-ingredient supplementation provides an additional nutritional stimulus to promote skeletal muscle anabolic responses to exercise with a moderate increase of caloric intake, which may actually assist older individuals in meeting their total daily energy intake requirements. Thus, for physically active, healthy, middle-aged, and older individuals, adding a preworkout multi-ingredient admixture, as used in our study, could be an acceptable strategy to maximise training adaptations and counteract the progression of age-associated declines in muscle mass, strength, and physical function.

The multi-ingredient admixture used in the present study provided 400 mg of anhydrous caffeine (5.2 ± 0.8 mg·kg^−1^ BM) and 300 mg of yerba mate extract (3.9 ± 0.6 mg·kg^−1^ BM), with an estimated per dose content of caffeine of about 1% (3 mg) (0.04 ± 0.01 mg·kg^−1^ BM) [49]. Thus, the resulting mean relative dose of caffeine per intake was 5.2 ± 0.8 mg·kg^−1^, which was within the range of recommended moderate doses (3 to 6 mg·kg^−1^) related to performance increase in resistance exercises [50]. In addition, the dose used was also slightly higher than the relative amount of caffeine included in preworkout supplements previously reported to be effective for improving resistance training volume [4,37,38].

Regarding the impact of preworkout supplementation on low-intensity endurance exercises, in line with the study of [5], those under the PREW treatment elicited higher fat oxidation. Nonetheless, different from the study of [5], who observed improvements in perceived response, our participants reported similar perception of global effort after both endurance and resistance training workouts. Discrepancies between studies may be due to differences between the supplements. Alkhatib and coworkers used a thermogenic-based multi-ingredient admixture with no protein, amino-acids, or carbohydrates, instead containing a combination of different herbs, i.e., 210 g of green tea leaf and 300 mg of Guarana seed extract, as well as 150 mg anhydrous caffeine per intake. The effects of this admixture were compared to those of a noncaloric placebo. Our study compared an admixture including thermogenic compounds (caffeine and yerba matte) protein, amino-acids, and carbohydrates vs. an iso-energetic supplement containing nonprotein or derivatives, or any thermogenic compound ingested prior to a low-intensity, middle-length (30 min) endurance exercise. The ingestion of macronutrients (e.g., carbohydrates) with added caffeine has been shown to reduce the perception of effort at the end of long duration or exhaustive endurance exercises [40,51]. Our participants were following typical eating patterns, having the last preworkout meal ~3-h before each exercise session. Therefore, it is unlikely that they were exercising with low muscle glycogen content or obtaining any advantage from the preworkout supplementation in terms of attenuating the perception of effort after performing a 30 min, submaximal endurance training session.

Since it is not possible to identify the contribution of each individual ingredient to the observed effects, several previous isolated nutrient studies have offered insight into the potential effects of the various nutrients [52]. In addition to the caffeine and yerba mate, the increased volume of weight lifted during RT may also be attributed to the improved vasodilatory response and higher muscular efficiency elicited by the ingestion of citrulline-DL-malate [53] and L-Carnitine [16]. Furthermore, the inclusion of 2 g per dose of betaine hydrochloride could have acted as an osmolyte, thereby increasing the water retention of cells and attenuating fatigue as the workout sessions progressed [54].

It has been claimed that the addition of theanine and tyrosine to caffeine favours cognitive function and focus on the task [55]. Although no effect was observed on the measured subjective feelings in this study, it is possible that these three ingredients worked synergistically with citrulline-malate, L-carnitine, amino-acids, and betaine to increase exercise volume by promoting cell hydration, blood flow, and the removal of metabolic by-products while attenuating the concomitant increase of the perception of effort due to the higher training volume. The dose of tyrosine used in the present investigation (1 g) was similar to the amount included in several multi-ingredient formulae claiming to enhance some aspects of exercise performance [4,56]. Nonetheless, to the best of the authors’ knowledge, there is still a lack of research on the synergistic effect of caffeine, thionine, and tyrosine to enhance muscular activation during exercise sessions.

The macronutrient admixture provided 14 g of isomaltulose (low glycaemic index ~32) and 1.9 g of maltodextrin (high glycaemic index > 90). Compared to ingesting maltodextrin only, the addition of isomaltulose to the protein and amino-acids admixture may have favoured a more stable glucose concentration [18], promoting a higher exercise volume during resistance training [57].

The present study had several limitations that must be considered when attempting to draw evidence-based inferences. Even though dietary records were kept, the participants’ diets were not fully controlled outside of the supplement routine. Although this approach has been extensively used, providing a prepacked diet to participants before and during the intervention would have offered a more accurate reliable means of achieving dietary control [58]. As several previous investigations have demonstrated the effectiveness of preworkout supplements on increasing performance outcomes vs. noncaloric conditions, the inclusion of another group receiving a nonenergy placebo supplement was not considered in the present investigation. Even though we allowed adjustment of the load during RT, variations of exercise intensity were not permitted during END. Thus, it was not possible to know whether the participants would have performed either a higher volume or maintained a higher exercise intensity with a similar rate of fat oxidation under the PREW condition compared to the CHO condition. Additionally, the supplementation protocol considered the absolute dose recommended by the manufacturer. Lastly, female participants were tested randomly over the menstrual cycle. The phase over which each participant was tested may influence individual strength performance [59] and or willingness to train [60].

Future studies should consider both acute and long-term interventions using individualised doses based on the participants’ body mass or fat-free mass. Regarding females, further studies analysing the impact of exercising in different phases of the menstrual cycle under diverse nutritional intervention are warranted.

## 5. Conclusions

The present investigation advocates for the ingestion of a preworkout, protein-based, multi-ingredient admixture providing ~5.2 mg·kg^−1^ of caffeine, 16 g (~0.21 g·kg^−1^) of carbohydrate including a high proportion of isomaltulose (slow-release disaccharide), and 9 g (~0.12 g·kg^−1^) of protein including added amino-acids (~1.8 ratio of CHO/protein), instead of carbohydrates alone, to increase resistance training volume and possibly enhance fat oxidation during endurance training in middle-aged, physically active men and women.

## Figures and Tables

**Figure 1 nutrients-12-03778-f001:**
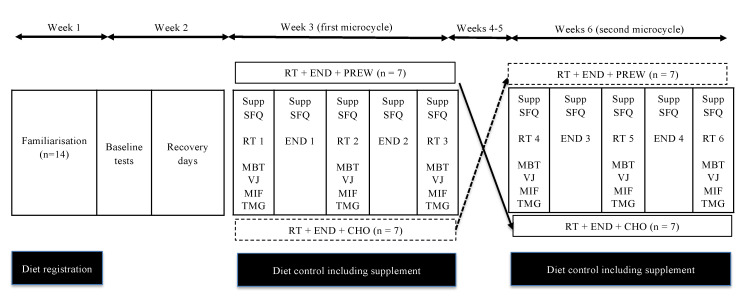
Overview of the study design. The overall intervention involved six consecutive weeks. First week: Familiarisation; Second week: baseline tests followed by a recovery period; Third week: First microcycle including three resistance workouts (RT 1, RT 2 and RT 3) and two endurance workouts (END 1 and END 2), pre- and post- training assessments; Fourth and Fifth weeks: Recovery/washout period; Sixth week: second microcycle including three resistance workouts (RT 4, RT 5 and RT 6) and two endurance workouts (END 3 and END 4), pre- and post- training assessments. **Supp**: Ingestion of the corresponding supplement condition [Preworkout multi-nutrient (PREW) or carbohydrate (CHO)]; **SFQ**: Participants completed the subjective feeling questionnaire; **MBT**: medicine ball throw test; **VJ**: vertical jump test; **MIF**: maximal isometric force test; **TMG**: Tensiomyography of vastus medialis, biceps femoris long head and anterior deltoids.

**Figure 2 nutrients-12-03778-f002:**
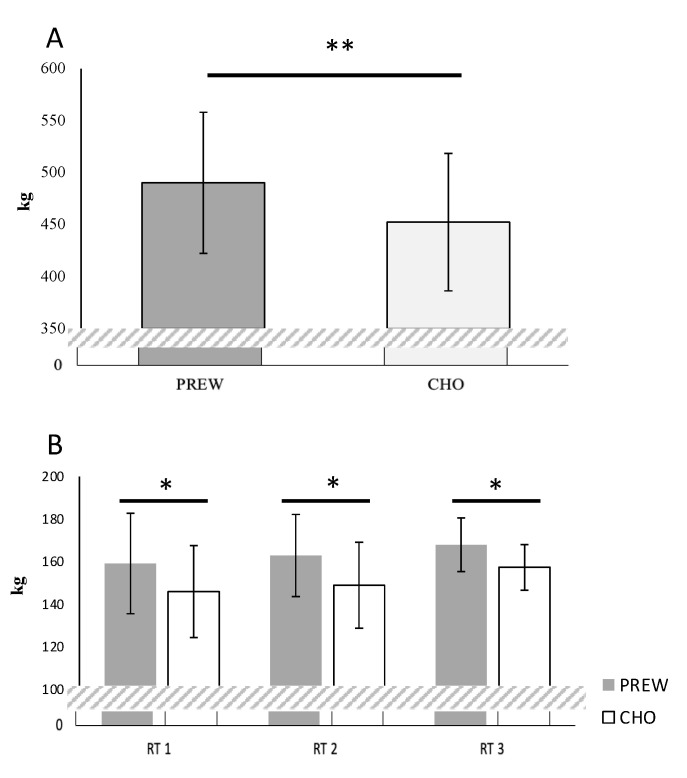
Estimated marginal mean values and 95% confidence intervals in resistance training volume (total kg lifted) per week (**A**) or per workout (**B**) under both treatment conditions: preworkout (PREW) or carbohydrate (CHO). ** *p* < 0.01, * *p* < 0.05 between conditions (PREW vs. CHO).

**Figure 3 nutrients-12-03778-f003:**
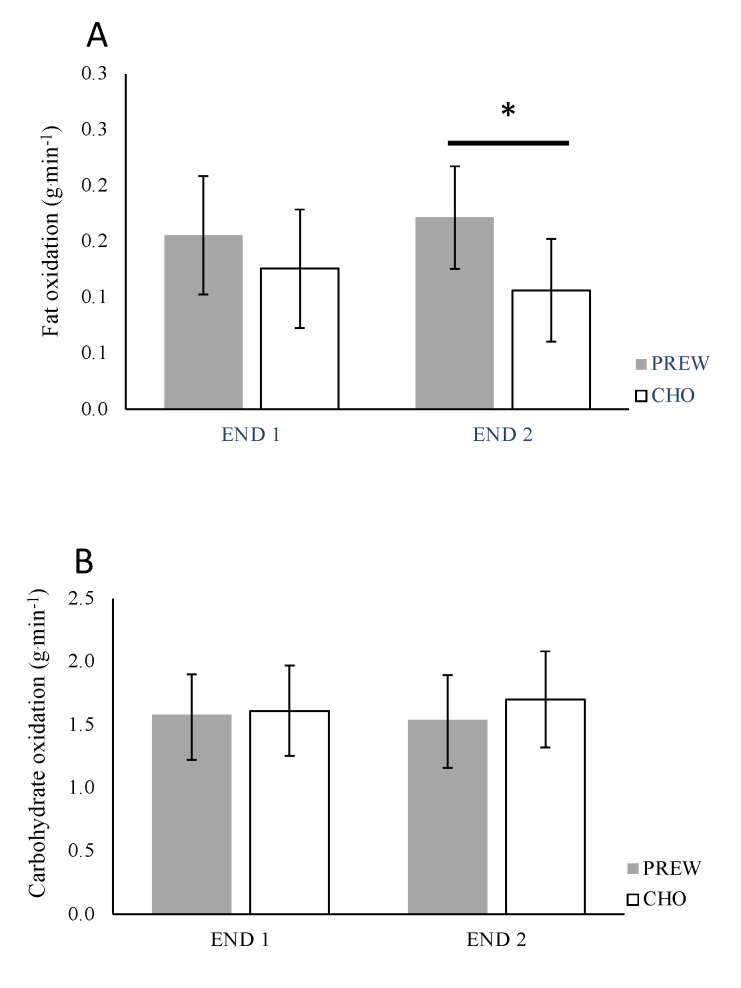
Estimated marginal mean values and 95% confidence intervals fat (**A**) and CHO (**B**), oxidation rate g·min^−1^ determined during the first (END 1) and second (END 2) endurance training session under both treatment conditions: preworkout (PREW) or carbohydrate (CHO). * *p* < 0.05 between conditions (PREW vs. CHO).

**Figure 4 nutrients-12-03778-f004:**
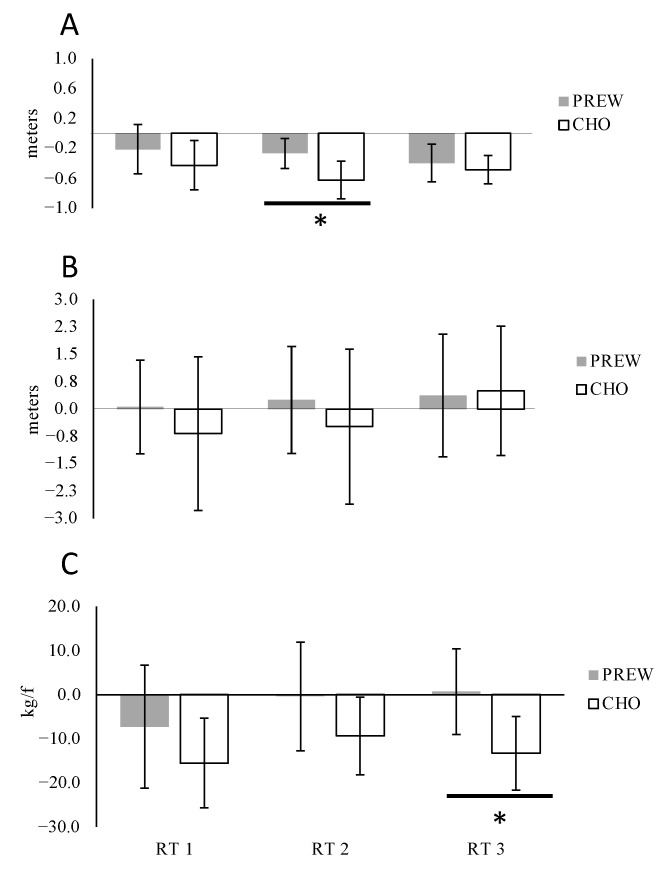
Estimated marginal mean values and 95% confidence intervals in delta changes in medicine ball throw (**A**), vertical jump (**B**), and maximal isometric force (**C**) measured after the completion of the three resistance training sessions (RT 1, RT 2 and RT 3) under both treatment conditions: preworkout (PREW) or carbohydrate (CHO). * *p* < 0.05 between conditions (PREW vs CHO).

**Table 1 nutrients-12-03778-t001:** Nutritional composition of supplements per intake mixing with ~350 mL of plain water.

Description	Multi-Ingredient (40 g dose)	Placebo (27 g dose)
Energy value (kcal)	100	102
**Macronutrients**		
Total carbohydrates (g) of which Isomaltulose (g) Maltodextrin (g)	~16 (14) (1.9)	25 (maltodextrin)
Total proteins included added amino acids (g)	9	-
**Amino acids and other ingredients**		
Betaine Hydrochloride (g)	2	-
L-Carnitine L-tartrate (g)	1.5	-
L-Citrulline-DL-malate (g)	2.5	-
L-Leucine (g)	3	-
L-Lysine (g)	2.7	-
L-Arginine Base (g)	2.5	-
L-Isoleucine (g)	1.5	-
L-Methionine (g)	0.7	-
L-Phenylalanine (g)	1.1	-
Taurine (g)	1	-
L-Threonine (g)	1.2	-
L-Tryptophan (g)	0.3	-
L-Tyrosine (g)	1	-
L-Valine (g)	1.5	-
Caffeine (mg)	400	
Yerba Mate extract (mg)	300	

**Table 2 nutrients-12-03778-t002:** Demographic characteristics of the participants (described by sex).

Measures	Males (n = 7) Mean ± SD (Range)	Females (n = 7) Mean ± SD (Range)
Age (yrs)	49 ± 5 (45–58)	49 ± 4 (45–55)
Height (cm)	178 ± 5 (168–185)	161 ± 3 (156–164)
Body mass (kg)	85.8 ± 14 (67–107)	71 ± 8 (58–82)
Fat-Free mass (kg)	63.3 ± 5 (52–69)	45.3 ± 5 (54–38)
Fat mass (kg)	25.1 ± 10 (12–38)	27.4 ± 9 (15–42)
Experience in RT (yrs)	2.4 ± 1 (1–5)	1.6 ± 1 (1–5)
Overhead Medicine ball Throw (m)	6.1 ± 1.2 (4.5–7.9)	4.6 ± 0.6 (3.6–5.4)
Countermovement jump (cm)	26.4 ± 3.4 (21.4–30.7)	17.1 ± 2.6 (12.9–20.4)
Maximal Isometric Mid-Thigh Pull (kgF)	179 ± 41 (114–240)	108.4 ± 18.8 (90.50–145.5)
$\dot{V}O_{2}$ peak (mL/kg/min)	47 ± 11 (34–71)	31 ± 5 (25–38)
Fat max intensity (Watts)	103 ± 45 (66–196)	55 ± 12 (41–74)

**Table 3 nutrients-12-03778-t003:** Descriptive analysis of participants diet compositions, including and not including preworkout supplementation.

Macronutrients	No Supplementation (n = 14)	With PreWorkout (n = 14)	With Maltodextrin (n = 14)
**Proteins**			
g·d^−1^	83.3 ± 31.4	92.9 ± 31.4 * ^φ^	83.3 ± 31.4 ^φ^
g·kg^−1.^d^−1^	1.1 ± 0.3	1.2 ± 0.33 * ^φ^	1.1 ± 0.3 ^φ^
% of total energy	16.5 ± 4.4	17.5 ± 4.24 * ^φ^	15.7 ± 4.2 * ^φ^
**Carbohydrate**			
g·d^−1^	242.0 ± 82.3	260.1 ± 83.1 * ^φ^	267.4 ± 82.4 * ^φ^
g·kg^−1.^d^−1^	3.2 ± 1.3	3.4 ± 1.3 * ^φ^	3.5 ± 1.3 * ^φ^
% of total energy	48.3 ± 13.3	49.0 ± 12.7 *	50.8 ± 12.7 * ^φ^
**Fats**			
g·d^−1^	77.4 ± 26.2	77.4 ± 26.2	77.4 ± 26.2
g·kg^−1.^d^−1^	0.97 ± 0.2	0.97 ± 0.2	0.97 ± 0.
% of total energy	35.2 ± 9.9	33.5 ± 9.4	33.4 ± 9.4
**Energy**			
Total daily energy	2053.2 ± 407.7	2167.4 ± 407.3 *	2157.3 ± 407.6 *
Kcal·kg^−1.^d^−1^	26.4 ± 4.9	27.9 ± 5.0 *	27.8 ± 5.0 *

* *p* < 0.01 to baseline ^φ^
*p* < 0.01 Preworkout vs. Maltodextrin. Notes: values are presented as mean ± standard deviation.

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
