# Peer review of "Effects of Multi-Ingredient Preworkout Supplementation across a Five-Day Resistance and Endurance Training Microcycle in Middle-Aged Adults"

_nutrients, 2020, doi:10.3390/nu12123778_

Round 1

Reviewer 1 Report

The present study aimed to compare the acute effects of ingesting a pre-workout multi-ingredient (PREW) over a training week (microcycle) including both resistance and endurance training vs. an isocaloric pre-workout placebo containing carbohydrate alone (CHO) in

recreationally trained middle-aged men and women on (i) resistance training performance (ii)substrate oxidation during endurance exercise (iii) post-workout decrease of muscle function, and (vi)

subjective measures.

Introduction

  1. Since a meta-analysis of 35 trials had already concluded that combining multi-ingredients with resistance training is an effective strategy to induce more significant gains of fat-free mass and strength, I am unclear about the unique contribution of this study?
  2. What are the research gaps this study attempted to fill-in?
  3. What are the health relevance and implications for the study?

Methods

  1. Unfortunately, the study's design does not allow the author to analyze and compare the ergogenic effect of this admixture. It was studied as a multi-ingredient as it is.
  2. Please clearly list the inclusion and exclusion criteria.
  3. Were subjects randomly assigned?
  4. What about the human subject protection protocol and procedure? Any IRB approval?
  5. How were the subjects compensated?
  6. How to assure that no supplements were consumed on non-exercising days? Any specific procedures for validation?
  7. For females, their menstrual cycles can affect exercise-induced fatigue and resistance training. Were these factors taken into consideration?

Results

  1. Although the findings are interesting, the very small sample size largely limits the generalizability of the results.

 Discussion

  1. Can the authors explain why there were no differences in all secondary outcomes?
  2. The authors should at least acknowledge some of the limitations.
  3. The discussion section is too brief. Authors should discuss the results and how they can be interpreted from previous studies and the working hypotheses.
  4. Future research directions may also be highlighted.

Reviewer 2 Report

Abstract:

  • Please, include the kind of design here.

Method:

  • Although this study is relevant, the sample size used is probably to small for performing such analysis, especially the ANOVA. Have the authors performed any pilot study in order to determine the appropriate sample size or performed power calculation for this particular study?
  • Inclusion and exclusion criteria need to be included.
  • How do you find the sample? This information is necessary to reply the study.
  • Including information about the age and the distribution in sexes of the participants.
  • Including information about the ethical committee and the inform consent of the participants.
  • Including information about who measured the variables (one or various researches), previous experience, qualification…
  • 1. Please, change the title. You have not measured only body composition.
  • Have do you did an ANOVA or a MANOVA? Furthermore, please, include the value for the p adjusted.

Results:

  • Please, included in table 2 t, p and effect size depends on sex. Include sex as covariable in the ANOVA and change the results according with this.
  • The results of ANOVA must include effect size. I propose you include a table after table 3 with the Inter-group differences (mean difference, 95%CI, p value and effect size).
  • No supplementation group are not included in figure 2, 3 and 4. Please, include it. Furthermore, can you included the 95%CI of these variables?

Discussion:

  • Lines 405-407: please, delete it and add here the discussion.

Section 5. Conclusions:

Please, include it before the conclusions paragraph.

Author Response

Dear Editor,

Thank you very much for considering our work for publication. We would like to extend our sincere thanks to the referees for the thorough review of our work and for their constructive comments. Based on the feedback, we have revised the manuscript and believe it will make a positive and required contribution to the literature in nutrients.

In recognition of the limitations of this study, we hope to have provided you with a publishable manuscript.

Responses to the reviewer 2

All points raised and suggestions made by the second reviewer have been seriously considered and responded on point-by-point basis

Abstract

Please, include the kind of design here

RESPONSE

A brief sentence describing the study design has been integrated into the abstract. It is important to highlight that further description is limited by the word-count of the abstract that is 200.

The following sentence has been edited as follows:

The present double-blind, cross-over study compared the acute effects of ingesting a pre-workout multi-ingredient (PREW) vs. carbohydrate (CHO) over a week (microcycle) composed by three resistance training (RT) workouts alternated with two 30-min low-intensity endurance sessions (END) on RT volume (kg lifted) and END substrate oxidation

Methods

Although this study is relevant, the sample size used is probably to small for performing such analysis, especially the ANOVA.

Have the authors performed any pilot study in order to determine the appropriate sample size or performed power calculation for this particular study?

Inclusion and exclusion criteria need to be included.

How do you find the sample? This information is necessary to reply the study.

Including information about the age and the distribution in sexes of the participants.

Including information about the ethical committee and the inform consent of the participants.

Including information about who measured the variables (one or various researches), previous experience, qualification…

RESPONSE:

Thanks you very much for your appreciations, we do apologise for missing this very important section in our first submission. The section 2.1 headed Participants is now included and describing the requested information.

2.1 Participants

Fourteen recreationally active, middle-aged adults (Seven females) participated in this study. To be eligible, participants required to have been training regularly 2 to 3 times per week, using routines including resistance exercises (e.g., bench press, leg press, squats, or lunges) for a minimum of 6 months before the beginning of the study. Exclusion criteria included anyone suffering from recent (last six months) or present injuries which may prevent them from performing the required exercises, suffering from current illnesses or chronic diseases (including metabolic syndrome, advanced obesity or sarcopenia), or taking any medication or supplements that would affect exercise performance (i.e., protein amino-acids supplements, NSAIDs, etc.). All female participants were pre-menopausal and were randomly tested throughout their menstrual cycle (MacNutt et al., 2012). All participants provided written informed consent in accordance with the Declaration of Helsinki. Procedures were approved by the University of Greenwich Research Ethics committee (FES-FREC-18-3.04.16) on 23rd January 2020. The project was registered as a clinical trial at the U.S. National Institutes of Health. https://www.clinicaltrials.gov (NCT041477741).

To determine the appropriate sample size, an interim analysis was performed once six participants completed the study. The analysis was conducted based on the most relevant primary outcome measure [the WVOL, summarizing the total load from the performed exercise over the three conducted resistance training sessions]. Assuming an α-error of 0.05, for the resulted effect size of d=0.96 calculated between two dependent means determined for the PREW and CHO conditions, the required sample size of n=11 was estimated to achieve 80% statistical power.

Please, change the title. You have not measured only body composition.

RESPONSE

With all respect to the reviewer, we are confused with the above question, we haven’t referred to body composition in our title. Indeed, although measurement of body mass, fat and fat-free mass were conducted, these data have been collected only for descriptive purposes.

Have do you did an ANOVA or a MANOVA? Furthermore, please, include the value for the p adjusted.

RESPONSE

As describe in section 2.5. Statistical analysis, based on each variable we conducted different statistical treatments, but in any case no MANOVA was used. For instance, a “two-way repeated-measure ANOVA [3 RT workouts or 2 END workouts  2 conditions (PREW vs CHO)] was performed to respectively analyse (i) the SVOL lifted and the session-RPE rated in each RT workout and (ii) the substrate oxidation (FAO and CHOox) and the averaged Borg-scale score measured during each END session”.

In the case of the second variables (e.g. performance measurements and TMG) in which differences from baseline were analysed at three time points after RT, as described “raw changes in performance (MBT, CMJ and MIF) and TMG, were calculated by subtracting pre from post-assessment values, without adjusting for pre-values, since the same participants performed under both conditions acting as their own controls. In order to assess the magnitude of the differences from baseline, confidence intervals (CIs) of the differences were calculated and plotted. Those CIs not crossing zero were considered statistically significant from the baseline performance. Additionally, two-tailed one-sample Student T-tests were used to test for a null effect hypothesis.

To compare differences between conditions (PREW vs CHO) at post-workout measurements in raw change, an ANOVA with repeated measures was used to examine changes over the 3 times (RT1, RT2 and RT3) for MBT, CMJ, MIF, VOL, GRPE (global rate of perceived exertion after 20 minutes of the end of the workout) and all TMG variables. Differences over time were compared using Bonferroni-adjusted pairwise comparisons when appropriate”.

Results:

Please, included in table 2 t, p and effect size depends on sex. Include sex as covariable in the ANOVA and change the results according with this.

RESPONSE

With all respect to the reviewer, table 2 is just a descriptive of the demographic characteristics of the participants. There is not any, t, p and ES value to include.

In addition, as described in the method section, “Previous analysis using sex as an inter-participant factor (i.e., condition  time  sex) demonstrated no significant interactions between sex and conditions or times for all the variables but TMG. Therefore, sex (men, women) was used as a covariate to analyse changes in the TMG variables. For the rest of the variables, data were pooled between sexes and analysed together for the rest of the variables”.

Data in table S1, submitted as supplementary material are presented using sex as co-variant.

The results of ANOVA must include effect size. I propose you include a table after table 3 with the Inter-group differences (mean difference, 95%CI, p value and effect size).

RESPONSE

Thank you very much for your suggestion, all relevant results observed in the primary outcomes have been presented in figures 2 and 3. Regarding the secondary outcomes, those observed on the performance variables showing some significant differences have also been presented in figure 4 (a, b, and c) while those of the TMG, demonstrating no significant differences to baseline nor between conditions have been described in Table S1 and submitted as supplementary material. Table S1 includes all the observed result of TMG variables using sex as co-variate, no sig differences nor moderate d>0.2 or large effect size d>0.8 were observed, therefore from our point of view there is no need to include these results.

All figures describe means and 95% CIs and relevant observed, moderate and large effect (d) are reported in the text.

With all respect to the reviewer, we consider that duplicating information provided in figures 1,2, 3 and 4 by the inclusion of another table describing the same results would not help readers to better understand our results. Nonetheless, we are opened to provide this data as supplementary material whether the editor considers this piece of work essential for publication.

No supplementation group are not included in figure 2, 3 and 4. Please, include it. Furthermore, can you included the 95%CI of these variables?

RESPONSE

With all respect to the reviewer, int our design there was no no-supplement group, we compared the impact of two different conditions in one singular group that was tested using CHO or PREW. Therefore, we cannot add a non-supplement group on figures 2, 3 and 4. On the other hand, we do apologise for not being clear enough with the information included in the figures’ legend, the bars in fact represent 95% CI. It has been now clarified.

Discussion:

Lines 405-407: please, delete it and add here the discussion.

RESPONSE

Sorry, when we were copying the original work into the nutrients template, we missed deleting some sentences. We do apologise for the confusion.

Section 5. Conclusions: Please, include it before the conclusions paragraph.

RESPONSE

Thanks for your observation, suggestion accepted.

REFERENCES

MacNutt, M. J., De Souza, M. J., Tomczak, S. E., Homer, J. L., and Sheel, A. W. (2012). Resting and exercise ventilatory chemosensitivity across the menstrual cycle. Journal of Applied Physiology, 112(5), 737–747. https://doi.org/10.1152/japplphysiol.00727.2011

Round 2

Reviewer 1 Report

Please make sure to include responses/changes in the manuscript by highlighting where (page number and paragraph #) they are located.